# Molecular Analysis of Vietnamese Patients with Mucopolysaccharidosis Type I

**DOI:** 10.3390/life11111162

**Published:** 2021-10-30

**Authors:** Ngoc Thi Bich Can, Dien Minh Tran, Thao Phuong Bui, Khanh Ngoc Nguyen, Hoang Huy Nguyen, Tung Van Nguyen, Wuh-Liang Hwu, Shunji Tomatsu, Dung Chi Vu

**Affiliations:** 1Vietnam National Children’s Hospital, 18/879 Lathanh, Dongda, Hanoi 100000, Vietnam; ngocctb@nhp.org.vn (N.T.B.C.); dientm@nch.gov.vn (D.M.T.); Buiphuongthao@hotmail.com (T.P.B.); khanhnn@nhp.org.vn (K.N.N.); 2Institute of Genome Research, Vietnam Academy of Science and Technology, 18-Hoang Quoc Viet, Hanoi 100000, Vietnam; nhhoang@igr.ac.vn (H.H.N.); tungnv@igr.ac.vn (T.V.N.); 3Department of Pediatrics, National Taiwan University Hospital, 8 Chung-Shan South Road, Taipei 10041, Taiwan; hwuwlntu@ntu.edu.tw; 4Nemours/Alfred I. duPont Hospital for Children, DuPont Experimental Station, Bldg. E400. #5205, 200 Powder Mill Rd., Wilmington, DE 19803, USA; Tomatsu@nemours.org

**Keywords:** α-L-iduronidase (IDUA) gene, IDUA gene mutation, mucopolysaccharidosis type I

## Abstract

Mucopolysaccharidosis type I (MPS I) is a rare autosomal recessive disorder caused by deleterious mutations in the α-L-iduronidase (*IDUA*) gene. Until now, MPS I in Vietnamese has been poorly addressed. Five MPS I patients were studied with direct DNA sequencing using Illumina technology confirming pathogenic variants in the *IDUA* gene. Clinical characteristics, additional laboratory results, and family history were collected. All patients have presented with the classical characteristic of MPS I, and α-L-iduronidase activity was low with the accumulation of glycosaminoglycans. Three variants in the *IDUA* gene (c.1190-10C>A (Intronic), c.1046A>G (p.Asp349Gly), c.1862G>C (p.Arg621Pro) were identified. The c.1190-10C>A variant represents six of the ten disease alleles, indicating a founder effect for MPS I in the Vietnamese population. Using biochemical and genetic analyses, the precise incidence of MPS I in this population should accelerate early diagnosis, newborn screening, prognosis, and optimal treatment.

## 1. Introduction

The mucopolysaccharidoses (MPS) are a group of inherited metabolic disorders caused by the deficiency of lysosomal enzymes that degrade glycosaminoglycans (GAGs). Although MPS disorders are distributed worldwide, there are regional differences in their distribution. Almost half of the patients in Vietnam with MPS have MPS II, while MPS I is rare [1]; the same distribution of MPS subtype is similar for other locations in Asia, such as Malaysia [2] Japan, Korea and Taiwan [3]. In contrast, the incidence of MPS I is higher than that of MPS type II in Western countries [3]. MPS I is caused by a deficiency of the *α*-L-iduronidase *(IDUA)*, which leads to an accumulation of two glycosaminoglycans (GAGs), heparan sulfate (HS) and dermatan sulfate (DS), in almost all tissues. MPS I is a progressive multisystemic disorder with a wide range of clinical manifestations. These include coarse facial features, hepatosplenomegaly, dysostosis multiplex, severe arthropathy, visual impairment, hearing loss, restrictive lung disease, upper airway obstruction, valvular heart disease, communicating hydrocephalus, mental retardation, and spinal cord compression [4]. MPS I is an autosomal recessive disease with an incidence of 1 in 100,000 live births [5] and is caused by mutations in the gene encoding alpha-L-iduronidase located on chromosome 4p16.3. The deficient IDUA disrupts the sequential degradation of the GAGs, DS and HS, resulting in lysosomal accumulation and excessive urinary excretion of partially degraded DS and HS [6]. The MPS I diagnosis is made clinically then confirmed by demonstrating deficient α-L-iduronidase activity in leukocytes or cultured cells. The gene encoding α-L-iduronidase (*IDUA*; MIM# 252800) contains 14 exons; the cDNA open reading frame (ORF) is ~2 kb in length and encodes a polypeptide of 653 amino acids [7]. Until now, 315 mutations in the *IDUA* gene have been reported (Human Gene Mutation Database; http://www.hgmd.org, accessed on 20 February 2021) [8]. Identifying specific variants in affected individuals is useful for precise heterozygote identification, prenatal diagnosis, and genotype-phenotype correlations. The proportion of patients with MPS type I in the north of Vietnam was 9.9% in a total of 71 MPS patients identified [1]. However, there has been no detailed report describing MPS I patients in the North of Vietnam. Here, we describe the clinical characteristics and molecular genetics of 5 patients with MPS I. 

## 2. Materials and Methods

### 2.1. Patients, Clinical Data and Biochemical Profiles

In this study, five patients were recruited to the Vietnam National Children’s Hospital. All of them are Kinh, the predominant ethnic group in Vietnam and four of them are from the same village, one commune, Hoai Duc district in the West of Hanoi, Vietnam. Clinical phenotype and biochemical tests were performed. Data included family history, sex, date of birth, gestational age, birth weight, date of onset the first symptoms, clinical course, and assessment of clinical features including coarse facial features, hepatosplenomegaly, dysostosis multiplex, severe arthropathy, visual impairment, hearing loss, upper airway obstruction, valvular heart disease, mental retardation, and spinal cord compression, inguinal or umbilical hernia, and dermal melanocytosis.

The patients underwent chest, long bone, spinal and skull X-rays, audiometry, optometry, ophthalmoscopy, and abdominal ultrasound in Vietnam National Children’s Hospital.

They were enzymatically diagnosed by demonstrating deficient IDUA activity in leukocytes, fibroblasts, cultured trophoblasts, and or dry blood spots, and GAG analysis was performed in fresh or dry urine spots in National Taiwan University Hospital. 

Urine GAG quantification, two-dimensional electrophoresis, and tandem mass spectrometry (MS/MS were performed for predominant disaccharide units of urinary GAGs. Leukocyte pellet was isolated from EDTA blood and used for fluorescent enzymatic assay of IDUA [9].

The activity of α-L-iduronidase enzymes measurement from dried blood spot samples was carried out on the SEEKER™ System using the NeoLSD assay.

### 2.2. Molecular Genetic Analysis

Targeted next-generation sequencing and variant confirmation were performed at the Invitae, San Francisco, USA. Reads are aligned to reference sequence GRCh37. The effect of intronic variants on splicing was analyzed by using Spliceman [10], ASSP [11], and Netgen2 [12]. The 3D model was built by using Swiss-PdbViewer software v.4.1.0 based on the published structure on the PDB protein bank with accession code 4MJ2 [13].

## 3. Results

### 3.1. Clinical Characteristics

A total of five patients were diagnosed with Hurler syndrome. Four patients came from the same small area in the Hoai Duc district. All patients were identified with variant of uncertain significance of *IDUA* gene (Table 1). Median ages at symptom onset and MPS I diagnosis are presented in Table 1. The average age at symptom onset for these patients was under 24 months, but the diagnosis was later at 27.1 ± 18.4 months.

Coarse facial features and dermal melanocytosis were the most predominant characteristic for all patients (Figure 1). In terms of musculoskeletal abnormalities, kyphosis/gibbus was present early in patients 3 and 4 at the age of 4 and 9 months, respectively. 

Hepatomegaly and long bone deformation were not present in all patients, but joint contractures were present in patient 3 and 5 at the 4 and 24 months of age, respectively. 

Concerning the thorax, the primary abnormality concerns the ribs, which can be “paddle-shaped” or “oar-shaped” (Figure 2). Modifications of the shape of vertebral bodies was saw in all patients, resulting in flattened and rounded vertebrae (Figure 3). At the thoraco-lumbar level, the vertebral body can show a deficiency in its anterosuperior corner and, consequently, an apparent prolongation of the anteroinferior one, resulting in the lateral X-ray in an “anterior beaking” aspect. When hypoplasia of both anterior corners occurs, the vertebral body is wedge-shaped.

Cardiac valve abnormalities were observed in three out of five patients. 

### 3.2. Molecular Genetic Analysis

Of the five MPS I patients, three had homozygous variants in *IDUA*, and two had compound heterozygous variants (Table 1, Figure 4). Patients 1, 2, 3, and 5 come from the same small area, but were not known to be related. The c.1046A>G (p.Asp349Gly) variant was reported as having unknown significance. Asp349 is located in a triosephosphate isomerase (TIM) barrel active site, interacting with substrate and playing an important role in the activity of IDUA. It has been reported that mutations at this residue (Asp349Tyr) will lead to the inactivation of the IDUA enzyme [14]. The c.1862G>C (p.Arg621Pro) variant, also reported as having unknown significance, causes a substitution of a hydrophilic amino acid arginine for a neutral amino acid proline, resulting in losses one out of three hydrogen bonds between Agr621 and Glu582 residues in the 3D model (Figure 5). The c.1190-10C>A variant occurs in the splice acceptor site of intron 8. The Spliceman, ASSP, and NetGene2 tools all predict that c.1190-10C >A has an 83% chance to shift the splicing position eight nucleotides upstream of the original site, resulting in truncation of the protein (Figure 6). 

## 4. Discussion

MPS I is a rare genetic disorder characterized by a broad spectrum of diseases with variable ages of onset, progression, and organ involvement. 

MPS I has been delineated into three different diseases based on clinical presentation, that is, Hurler syndrome (severe), Hurler Scheie syndrome (intermediate), and Scheie syndrome (mild). Guidelines of the American Academy of Pediatrics recommend classifying the MPS I spectrum into two broader groups, that is, severe MPS I (Hurler syndrome) and attenuated MPS I (Hurler Scheie and Scheie syndromes) [15].

Hurler syndrome, the most severe form of MPS I, typically involves significant developmental delay and cognitive decline, along with characteristic coarse facial features, joint stiffness and contractures, short stature, and respiratory, cardiac, and hepatic disease. Patients with Hurler syndrome were diagnosed earliest compared with Hurler Scheie and Scheie syndromes [16]. In a study by Beck et al. [5], the median age at symptom onset for patients with Hurler syndrome was 6 months, and diagnosis followed quickly after that, at median ages of 12 months. In our study, the average age at symptom recognition and diagnosis age was later (Table 1). Early symptom recognition facilitates early diagnosis in patients with MPS I. However, due to the rarity of the disease and the variability of clinical manifestations, MPS I poses challenges for diagnosis, for parents and healthcare providers. Correct diagnosis of MPS I in a child is necessary to provide treatment for the individual, and importantly, provide the family with genetic counseling regarding the implications of the diagnosis for their child and recurrence risk for subsequent children. 

The nucleotide change c.1046A>G (p.Asp349Gly) was identified in patients 3 and 5. This variant has been reported on the dbSNP database with the ID rs371397270. Asp349 is a binding site on the *IDUA* amino acid sequence and plays an important role in the activity of IDUA. Other variants that disrupt this residue have been observed in individuals with *IDUA*-related conditions [17,18], suggesting that this may be a clinically significant amino acid residue. Some mutations that occurs in residue 349 have beenpreviously been reported in MPS I patients including Asp349Tyr [14] and Asp349Asn [19]. Both of them produced high levels of catalytically inactive α-L-iduronidase protein. Therefore, this variant may lead to a decrease or loss of catalytic activity of IDUA [19]. This variant is on the list of 91 nsSNP predicted as damaging by SIFT, PolyPhen, I-Mutant, PROVEAN and on the two lists of 28 nsSNP predicted as associated with disease by PHD-SNP, PANTHER, and SNP&GO [19].

The variant c.1862G>C (p.Arg621Pro) occurs in exon 14 of the *IDUA* gene, causing a substitution of a hydrophilic amino acid Arginine for a neutral amino acid Proline. The p.Arg621Pro variant loses one out of three hydrogen bonds between Agr621 and Glu582 residues. This variant has not been reported in the ExAC database. Many tools, including Provean, have predicted the effects of this variant on protein functions: “Deleterious”; Mutation Taster: “Disease-causing”; SIFT: “Deleterious”; PolyPhen-2: “Probably Damaging”; Align-GVGD: “Class C0”; however, they all failed to provide any consistent results.

The variant c.1190-10C>A occurred in the eighth intron region of the *IDUA* gene. This variant has been identified in homozygous or heterozygous forms in four patients who were born and live in the same area. Although the intron regions are not involved in encoding proteins, mutations in these regions can still negatively impact the protein formation by altering the pre-mRNA binding site [20,21,22]. The shifting of splicing position would likely add eight extra nucleotides from the intron region to the mature mRNA sequence, resulting in a frameshift that gains a stop codon after 45 amino acids downstream of the mutation site, leading to truncated 212 amino acids. Many of the variants of the *IDUA* gene have been described [7,23]. It has been shown that variants can alter the patient’s clinical phenotype [24]. Therefore, identification of the mutation spectrum in *IDUA* gene could be beneficial for further research of MPS I in Vietnamese populations as well as supporting the clinical testing and diagnosis of MPS I. This study found that the c.1190-10C>A variant is associated with the Hurler phenotypes.

## 5. Conclusions

We have identified the putative founder mutation, c.1190-10C>A from four of the five patients derived from the same village. Further haplotype analysis is required to prove the founder effect of c.1190-10C>A, however it is hindered because of the small size and number of the families carrying the mutation. Prevalence of this variant in other areas of the country will also be important to explore the incidence, newborn screening, prognosis, and optimal treatment of MPS I in Vietnam.

## 6. Limitation of the Study

We have described five MPS I patients with the same variant unreported. Although the incidence of MPS I is rare, we have encountered five patients in unrelated families from the same small community with the same variant. This variant will be responsible for MPS I disease in our patients. Although we have not proven the functional assay of this variant. The haplotype analysis proves the founder effect at the molecular level. However, we could not perform the haplotype analysis. We had four patients who presented a previously not described intronic mutation required to confirm the splicing impact of this variant with mRNA data. However, it was challenging to obtain RNA materials from the specimens of the patients. Therefore, we could not confirm it.

## Figures and Tables

**Figure 1 life-11-01162-f001:**
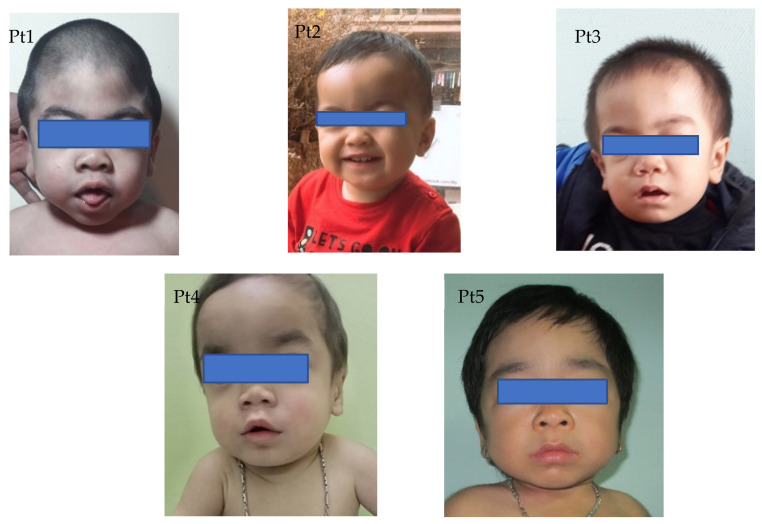
Photos of patients: Coarse face.

**Figure 2 life-11-01162-f002:**
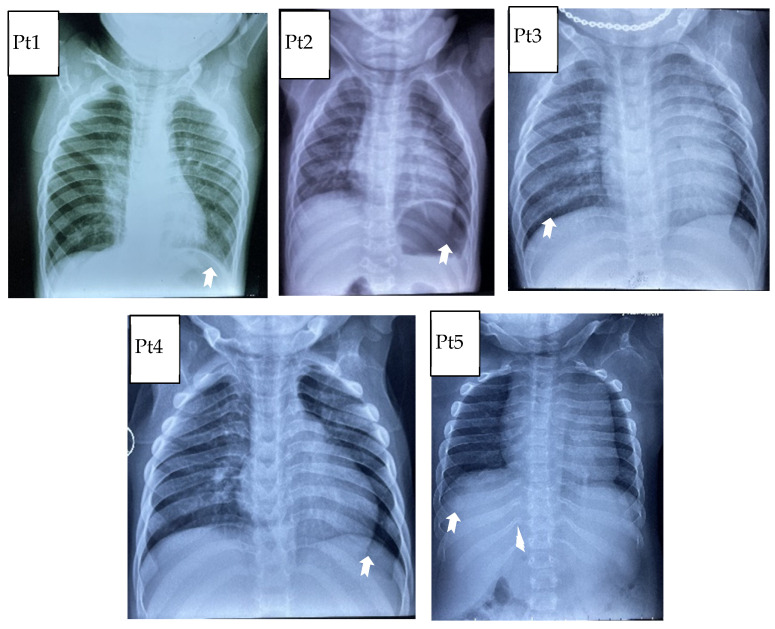
Thoracic abnormality: “paddle-shaped” or “oar-shaped” because of the widening of the anterior arches and of the tapering of the posterior ones.

**Figure 3 life-11-01162-f003:**
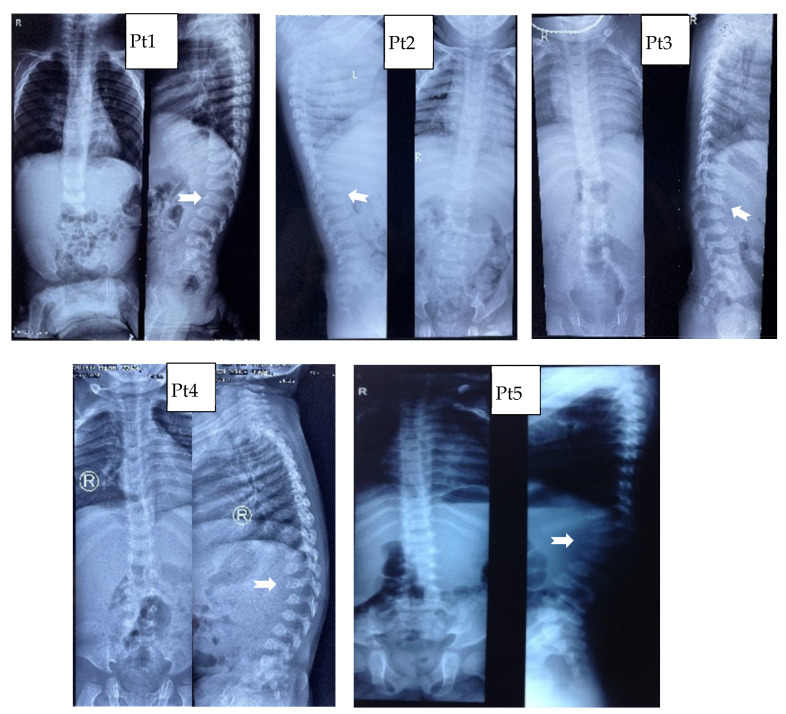
Spine abnormality: X-ray of multiplex dysostosis of the spine, vertebral bodies rounded, The “anterior beaking” aspect with posterior scalloping and the platyspondylia with “wedge-shaped” deformity.

**Figure 4 life-11-01162-f004:**
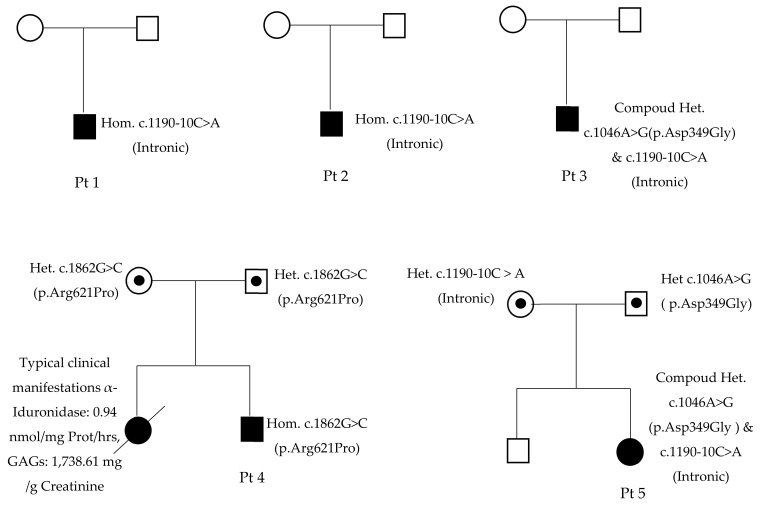
Pedigree of patients.

**Figure 5 life-11-01162-f005:**
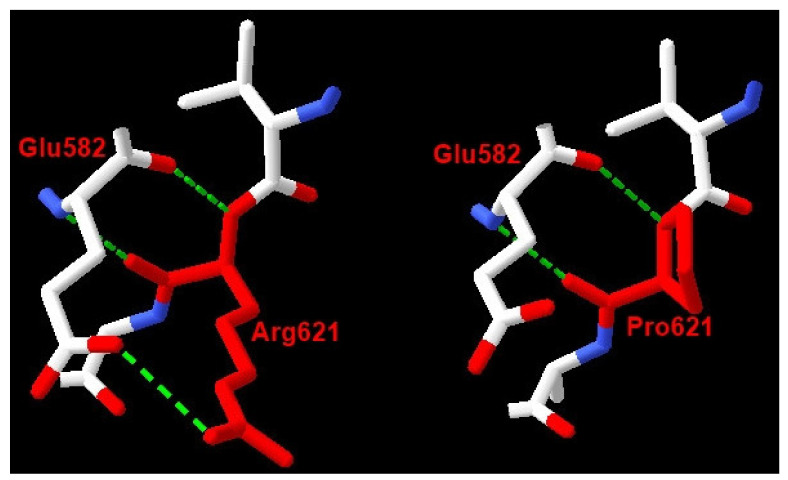
The 3D structure alteration was built based on the published structure on the PDB protein bank with accession code 4MJ2 by using Swiss-Pdb Viewe software v.4.1.0. Mutation causing change in number of hydrogen bonds between residues. Arg621 has 3 hydrogen bonds with Glu582 while Pro621 has only 2 hydrogen bonds with Glu582.

**Figure 6 life-11-01162-f006:**
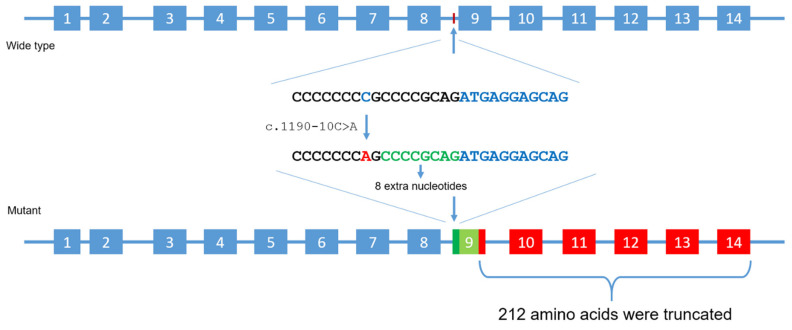
The c.1190-10C>A mutation may cause activation of an alternative cryptic splice site would likely add 8 extra nucleotides from the intron region to the mature mRNA sequence, resulting in a frameshift which gain a stop codon after 45 amino acids downstream of the mutation site, lead to truncated 212 amino acids.

**Table 1 life-11-01162-t001:** Clinical characteristics, molecular genetics of patients with MPS at diagnosis.

Patient	Pt1	Pt2	Pt3	Pt4	Pt5
Age of diagnosis (months)	20	15	38	9	54
Age onset of the first symptoms (months)	20	15	4	9	24
The first symptoms	cornea clouding	coarse face	kyphosis	kyphosis	Joint stiffness
Coarse face	+	+	+	+	+
Cornea clouding	+	−	+	−	+
Joint stiffness	−	−	+	−	+
Long bone deformation	−	−	−	−	−
Spinal deformation	−	−	+	+	−
Hearing loss	−	+	NA	+	+
Valvular heart disease	+	+	+	−	−
Inguinal or umbilical hernia	−	−	+	+	−
Dermal melanocytosis	+	+	+	+	+
Hepatosplenomegaly	−	−	−	−	−
α-L-iduronidase (dried blood spots, normal >1.32 µM/h)	0.17	0.06	0.13		
α-L-iduronidase (leukocytes, normal 15.9–41.8 nmol/mg prot/h)				0.01	0.43
Urine heparan sulfate (mg/mmol creatinine)	8.26 (<0.67)	55.44 (0.65–3.51)	8.18 (<0.76)		
Urine dermatan sulfate (mg/mmol creatinine)	11.89 (<0.24)	14.07 (0.14–0.90)	11.55 (<0.21)		
Urine keratan sulfate (mg/mmol creatinine)	0.32 (< 0.73)	2.08 (0.08–0.98)	0.23 (<0.92)		
Urine glycosaminoglycans (mg/g creatinine)				1106.4 (10.7–112)	508.8 (10.7–112)
Variant 1	c.1190-10C>A	c.1190-10C>A	c.1190-10C>A	c.1862G>C	c.1190-10C>A
Variant 2	c.1190-10C>A	c.1190-10C>A	c.1046A>G	c.1862G>C	c.1046A>G

‘’+’’: Yes; ‘’−’’: No; ‘’NA’’: Not analysis.

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
