# Peer review of "Molecular Analysis of Vietnamese Patients with Mucopolysaccharidosis Type I"

_life, 2021, doi:10.3390/life11111162_

Round 1
Reviewer 1 Report
This is a case report of 5 patients with MPS I. Clinical, biochemical and molecular data are presented.
Four patients present a previously not described intronic mutation, that in silico predictors consider altering splicing. It would be important if authors could confirm that it indeed alter splicing with mRNA data.
The other two mutations p.R621P and p.R349G are also classified as VUS but the additional data and discussion presented fails to convince readers that these variants could indeed be pathogenic. The discussion on p.R349G is rather confusing. Regarding p.R621P, other variants present in the same position were not even mentioned in the manuscript.
As patients were from a delimited geographic region, haplotype analysis could have added some insight into wether or not these variants have a common origin. There is a comment on that at the discussion, but no data is presented.
Author Response
Dear Sir/Madam,
Thank you very much for your comments. We appreciate your insightful comments. We have addressed all concerns and comments point-to-point as follows:
1. This is a case report of 5 patients with MPS I. Clinical, biochemical and molecular data are presented.
Response: We have described clinical, biochemical, and molecular data to prove these patients are diagnosed with Hurler syndrome.
2. Four patients present a previously not described intronic mutation, that in silico predictors consider altering splicing. It would be important if authors could confirm that it indeed alter splicing with mRNA data
Response: We agree that it is very important to confirm the splicing impact of this variant with mRNA data. However, it was challenging to obtain RNA materials from the specimens of the patients. Therefore, we could not confirm it. We have described this limitation in the text
3. The other two mutations p.R621P and p.R349G are also classified as VUS but the additional data and discussion presented fails to convince readers that these variants could indeed be pathogenic. The discussion on p.R349G is rather confusing. Regarding p.R621P, other variants present in the same position were not even mentioned in the manuscript
Response: We have corrected the text according to your comment, describing these two mutations in detail
4. As patients were from a delimited geographic region, haplotype analysis could have added some insight into whether or not these variants have a common origin. There is a comment on that at the discussion, but no data is presented
Response: Thank you very much for your critical comments. We agree that the haplotype analysis proves the founder effect at the molecular level. Since we could not obtain further samples, we could not perform the haplotype analysis. We have described this limitation in the text.

Reviewer 2 Report
In this study the authors report on the molecular characterization of five Vietnamese patients with mucopolysaccharidosis type I. Although the findings are of relevance in the clinical and epidemiological context of the country, the manuscript contains many inaccuracies and weaknesses that compromises its quality.
As so, I cannot recommend the ms for publication in this form, and I hope the authors will consider making the revisions I’m suggesting herein to give more accuracy and clarity to the current version.
1.In the abstract, (pag1, lines 17-18) saying that “more than 200 pathogenic variants have been described so far, but their frequencies have not yet been analyzed on a worldwide scale”, would suggest that the worldwide distribution of those variants would be addressed, which is far from the scope of the article. The second part of phrase must be eliminated or substituted by something informing that “ (variations) present a rather uneven distribution across worldwide populations”
- Pag1.lines 38-39. When referring to MPS I incidence, the correct description must be “an incidence of 1 in 100,000 live births”;
- Pag1.lines 42-44. Rephrase the sentence to something like “The primary clinical diagnosis of MPS I must be confirmed biochemically by demonstrating deficient α-L-iduronidase activity in leukocytes or cultured cells”
- Pag1.lines 44-45. Note that OMIM number for IDUA is OMIM*252800, not OMIM#252800 (hash instead of asterisk and OMIM instead MIM, assuming the online database was used);
- Pag 2 line 47. Whereas here is said that “So far more than 100 different disease-causing IDUA mutations have been reported (in HGMD)”, before, in the abstract, it is written “More than 200 pathogenic variants have been de-17 scribed so far”. Be more coherent on this number as well on the sources used to compute the estimates.
- Pag 2 line 50-51 The following statement needs improvement “Identifying specific mutations in affected individuals is useful for precise heterozygote identification, prenatal diagnosis, and genotype-phenotype correlations” Here goes a suggestion “Identifying specific mutations in affected individuals is important to obtain a precise molecular characterization, to perform prenatal diagnosis and genetic counselling, and to investigate genotype-phenotype relationships”
- The entire section Material and Methods needs high attention.
7.1. A proper 2.1 sub-title should be weighted. A hypothetical alternative could be “2.1. Patients, clinical data and biochemical profiles”
7.2. It must be better indicated to which ethnic group belong 4 of the patients. It is indicated they are “King ethenic… “, which, besides the typing mistake, needs to be confirmed. And what about the fifth patient? From where is he?
7.3. In point 2.1, information must be given on whether the study follows the current guidelines on ethic concerns (was it approved by any ethic committee) and on whether informed consent was obtained from the patients’ guardians;
7.4. Section “2.2. Mutation Analysis” also needs much improvement. Instead of listening possible technologies for confirming NGS results in general, which are widely known, a brief description of specific one used to confirm the variants in IDUA reported in the ms must be presented;
In this section, it is unnecessary to explain what is a MedGen ID or a OMIM number.
7.5. It must be indicated which Ensembl sequence of the IDUA gene was used as reference for residue numbering, and whether the mutation nomenclature followed the guidelines and recommendations of the Human Genome Variation Society.
7.6. The tools used to perform 3D modeling must be presented in Material and Methods, not in Results as the authors did.
- Pag 2 lines 90-91. Instead “All patients were Hurler syndrome, “, change, for instance, to “All patients showed clinical manifestations and symptoms of Hurler syndrome,…”
- pag 3 line 98. Avoid “Mongolian blue spots”. Nowadays those spots are referred to as dermal melanocytosis.
10-In section 3.1.2., when the variants are addressed individually, the discrimination of the genotype (homozygous or heterozygous) has no sense. Eliminate that information.
11- A discrepancy exist on the variants detected in patient #5: while in the Table 4 appears c.1190-10C>A (Intronic)/c.1046A>G (p.Asp349Gly); in Figure 2 appears c.1862G>C(p.Arg621Pro) & c.1190 10C>A. Check and correct that.
- Concerning the detected variants, my major concern is about c.1190-10C>A, which the authors assume to be a likely splicing mutation. However, evidence provided is unclear and globally weak. In the database ClinVar, there are entries to other alterations in c.1190-10 (c.1190C>T; c.1190-10dup; c.1190-10del), some of which were classified as benign. If the position affects an intronic polyC tract, it may be difficult to obtain an unambiguous detection. On the other hand, the authors did not analyzed cDNA, which could give more strength to the prediction. If cDNA is not available, at least further data must be presented on how the variation was clearly confirmed; the results obtained with the in silico tools must be more explored; and a more convincing explanation on how the variation affects splicing must be given.
- pag 6 lines 185-206. This first part of the Discussion, does not fit the aims of the section. The information presented should be reassigned to Introduction, complementing the questions already addressed. In the Discussion the authors must focus the patients studied, discussing their clinical characteristics in the broad context of MSPI
- pag 7, lines 240-on. The detection of c.1190-10C>A in homozygosity in two patients and compound heterozygosity in other two, needs to be more discussed. If all the patients belong to the same ethnic group, it would be interesting to comment a little more on the subject.
- There are numerous issues with the writing. The problems are so frequent that it is sometimes difficult to concentrate on the message and to understand the meaning. The manuscript needs strong improvement in the English writing.
Author Response
Dear Sir/Madam,
Thank you for your brilliant comments. We have revised the results section more clearly according to your suggestion. We have addressed all concerns and comments point-to-point as follows:
In this study, the authors report on the molecular characterization of five Vietnamese patients with mucopolysaccharidosis type I. Although the findings are of relevance in the clinical and epidemiological context of the country, the manuscript contains many inaccuracies and weaknesses that compromises its quality.
As so, I cannot recommend the ms for publication in this form, and I hope the authors will consider making the revisions I’m suggesting herein to give more accuracy and clarity to the current version.
Response: We have reviewed and edited the manuscript through the text. We hope that the current version will satisfy the reviewer and the readers
1. In the abstract, (pag1, lines 17-18) saying that “more than 200 pathogenic variants have been described so far, but their frequencies have not yet been analyzed on a worldwide scale”, would suggest that the worldwide distribution of those variants would be addressed, which is far from the scope of the article. The second part of phrase must be eliminated or substituted by something informing that “ (variations) present a rather uneven distribution across worldwide populations”
Response: We agree with your suggestion. We have added more information about uneven distribution across worldwide populations of Mucopolysaccharidosis
2. Pag1.lines 38-39. When referring to MPS I incidence, the correct description must be “an incidence of 1 in 100,000 live births”;
Response: According to your suggestion, we have corrected it.
3. Pag1.lines 42-44. Rephrase the sentence to something like “The primary clinical diagnosis of MPS I must be confirmed biochemically by demonstrating deficient α-L-iduronidase activity in leukocytes or cultured cells”
Response: We have rephrased this sentence to: “The MPS I diagnosis is made clinically, then confirmed by demonstrating deficient α-L-iduronidase activity in leukocytes or cultured cells”
4. Pag1.lines 44-45. Note that OMIM number for IDUA is OMIM*252800, not OMIM#252800 (hash instead of asterisk and OMIM instead MIM, assuming the online database was used);
Response: We have revised it
5. Pag 2 line 47. Whereas here is said that “So far more than 100 different disease-causing IDUA mutations have been reported (in HGMD)”, before, in the abstract, it is written “More than 200 pathogenic variants have been de-17 scribed so far”. Be more coherent on this number as well on the sources used to compute the estimates
Response: We have corrected it accordingly
6. Pag 2 line 50-51 The following statement needs improvement “Identifying specific mutations in affected individuals is useful for precise heterozygote identification, prenatal diagnosis, and genotype-phenotype correlations” Here goes a suggestion “Identifying specific mutations in affected individuals is important to obtain a precise molecular characterization, to perform prenatal diagnosis and genetic counselling, and to investigate genotype-phenotype relationships”
Response: Thank you for your critical comment. We have revised the text accordingly
7. The entire section Material and Methods needs high attention.
Response: We have revised it.
7.1. A proper 2.1 sub-title should be weighted. A hypothetical alternative could be “2.1. Patients, clinical data and biochemical profiles”
Response: We have corrected it.
7.2. It must be better indicated to which ethnic group belong 4 of the patients. It is indicated they are “King ethenic… “, which, besides the typing mistake, needs to be confirmed. And what about the fifth patient? From where is he?
Response: We have revised the text as suggested.
7.3. In point 2.1, information must be given on whether the study follows the current guidelines on ethic concerns (was it approved by any ethic committee) and on whether informed consent was obtained from the patients’ guardians;
Response: We have added more information in this regard.
7.4. Section “2.2. Mutation Analysis” also needs much improvement. Instead of listening possible technologies for confirming NGS results in general, which are widely known, a brief description of specific one used to confirm the variants in IDUA reported in the ms must be presented; In this section, it is unnecessary to explain what is a MedGen ID or a OMIM number.
Response: We have revised it.
7.5. It must be indicated which Ensembl sequence of the IDUA gene was used as reference for residue numbering, and whether the mutation nomenclature followed the guidelines and recommendations of the Human Genome Variation Society
Response: We have revised it.
7.6. The tools used to perform 3D modeling must be presented in Material and Methods, not in Results as the authors did.
Response: The tools used to perform 3D modeling were moved from the Results section to the Materials and Methods section
8. Pag 2 lines 90-91. Instead “All patients were Hurler syndrome, “, change, for instance, to “All patients showed clinical manifestations and symptoms of Hurler syndrome,…”
Response: We have revised the text as suggested: “A total of 5 patients were diagnosed with Hurler syndrome”
9. pag 3 line 98. Avoid “Mongolian blue spots”. Nowadays those spots are referred to as dermal melanocytosis.
Response: We have changed “Mongolian blue spots” to “dermal melanocytosis” as suggested.
10. In section 3.1.2., when the variants are addressed individually, the discrimination of the genotype (homozygous or heterozygous) has no sense. Eliminate that information.
Response: We have eliminated that information.
11. A discrepancy exist on the variants detected in patient #5: while in the Table 4 appears c.1190-10C>A (Intronic)/c.1046A>G (p.Asp349Gly); in Figure 2 appears c.1862G>C(p.Arg621Pro) & c.1190 10C>A. Check and correct that.
Response: We confirmed that patient 5 had two variants: c.1190-10C>A (Intronic) and c.1046A>G (p.Asp349Gly). We have revised it.
12. Concerning the detected variants, my major concern is about c.1190-10C>A, which the authors assume to be a likely splicing mutation. However, evidence provided is unclear and globally weak. In the database ClinVar, there are entries to other alterations in c.1190-10 (c.1190C>T; c.1190-10dup; c.1190-10del), some of which were classified as benign. If the position affects an intronic polyC tract, it may be difficult to obtain an unambiguous detection. On the other hand, the authors did not analyzed cDNA, which could give more strength to the prediction. If cDNA is not available, at least further data must be presented on how the variation was clearly confirmed; the results obtained with the in silico tools must be more explored; and a more convincing explanation on how the variation affects splicing must be given.
Response: Unlike other alterations in c.1190-10 which were classified as benign, c.1190-10C>A produced new “CAG” motif at acceptor-splice in the 8th intron. Therefore, it is more likely that the point mutation is to disrupt normal splicing. We have described it in more detail to convince the readers.
13. pag 6 lines 185-206. This first part of the Discussion, does not fit the aims of the section. The information presented should be reassigned to Introduction, complementing the questions already addressed. In the Discussion the authors must focus the patients studied, discussing their clinical characteristics in the broad context of MSPI
Response: We have deleted this paragraph.
14. pag 7, lines 240-on. The detection of c.1190-10C>A in homozygosity in two patients and compound heterozygosity in other two, needs to be more discussed. If all the patients belong to the same ethnic group, it would be interesting to comment a little more on the subject.
Response: We have revised this paragraph
15. There are numerous issues with the writing. The problems are so frequent that it is sometimes difficult to concentrate on the message and to understand the meaning. The manuscript needs strong improvement in the English writing.
Response: We have improved our manuscript edited by a native speaker
We would like to send you the revised manuscript using the “Track
Changes” function in the attached file.
Thank you very much,
Kind Regards,
Bich Ngoc

Reviewer 3 Report
Materials and methods section. It is not clear how IDUA enzyme activity, urine GAG level, heparan and dermatan sulfate level were measured.
Results section. It is not clear how the diagnosis of Hurler syndrome, instead of Scheie or Hurler-Scheie, was made. It is important to understand the rationale because a major claim of this manuscript is three mutations associated with Hurler syndrome.
Please be sure to use the correct nomenclature of mutations. Many studies in this field are using different nomenclatures, making it difficult to keep track of each mutation. There is a guideline, but I forgot its title.
Has any neurocognitive tests, e.g., Bayley, Vineland, been done?
If X-rays, audiometry, ultrasound have been performed, any representative data to share? These results are way more informative than Figure 3, to be frank.
Table 2. Why only patient 1, 2, and 4 have IDUA activity data in dry blood spots? Why only patient 3 and 5 have IDUA activity data in leukocytes?
Table 3. Similarly, why only some patients have heparan and dermatan sulfate data?
Table 3. Urine GAG is usually reported as ug GAG/mg creatinine.
It is not clear whether any new mutations were identified in this study. If yes, I would suggest to highlight it in the abstract.
Author Response
Dear Sir/Madam,
Thank you very much for your brilliant comments. We have addressed all concerns and comments point-to-point as follows:
Comments and Suggestions for Authors:
1. Materials and methods section. It is not clear how IDUA enzyme activity, urine GAG level, heparan, and dermatan sulfate level were measured.
Response: We have described it in detail.
2. Results section. It is not clear how the diagnosis of Hurler syndrome, instead of Scheie or Hurler-Scheie, was made. It is important to understand the rationale because a major claim of this manuscript is three mutations associated with Hurler syndrome.
Response: We have described the determination of clinical phenotype.
3. Please be sure to use the correct nomenclature of mutations. Many studies in this field are using different nomenclatures, making it difficult to keep track of each mutation. There is a guideline, but I forgot its title.
Response: We have corrected the nomenclature of mutation.
4. Has any neurocognitive tests, e.g., Bayley, Vineland, been done?
Response: We have used the Raven and Denver test in this study.
5. If X-rays, audiometry, ultrasound have been performed, any representative data to share? These results are way more informative than Figure 3, to be frank.
Response: We have added X-ray images of the chest and spine.
6. Table 2. Why only patient 1, 2, and 4 have IDUA activity data in dry blood spots? Why only patient 3 and 5 have IDUA activity data in leukocytes?
Response: We sent the whole blood samples and fresh urine to Taiwan to analyze the IDUA activity in leukocytes and urine GAG. However, after that, it is difficult to send the whole blood samples and fresh urine (liquid) to Taiwan, so that we have to the dry blood spots and dry urine spots samples instead.
7. Table 3. Similarly, why only some patients have heparan and dermatan sulfate data?
Response: Previously, we quantified total GAG in urine. Recently, we have quantified HS and DS in the specimens. Therefore, there is some difference in what we have analyzed. We have described it in the text.
8. Table 3. Urine GAG is usually reported as ug GAG/mg creatinine.
Response: We have corrected it according to the suggestion. Urine GAG was reported as mg/g creatinine.
9. It is not clear whether any new mutations were identified in this study. If yes, I would suggest to highlight it in the abstract.
Response: All variants are reported. We have suggested that these variants are novel and unreported in the abstract.
We would like to send you the revised manuscript using the “Track
Changes” function in the attached file.
Thank you very much,
Kind Regards,
Bich Ngoc

Round 2
Reviewer 1 Report
The authors have significantly improved their manuscript, although haplotype studies could have been performed with existing molecular data.
Author Response
Dear Sir/Madam,
Thank you very much for your critical comments. We agree that the haplotype analysis proves the founder effect at the molecular level. However, we could not perform the haplotype analysis. We have described this limitation in the text.
We would like to send you the newly revised manuscript in the attached file.
Kind Regards,
Bich Ngoc

Reviewer 2 Report
I thank the authors for having considered my main suggestions and comments on this reviewed version of the manuscript.
However, my major concern, which was about c.1190-10C>A, continues. The explanation on the possible deleterious effect of this variation is rather chaotic and unsatisfactory. In page 3 lines 1273-128 the authors write that “…changing from "agccccgcag^ATGAGGAGCA" to "ccccccccag^CCCCGCAGAT", resulting in truncation of the protein” First, it must be more clearly evidenced how that specific splicing disruption will led to a truncated protein. Second, it must be explained how the truncation of the protein arises, given that in lines 264/265 the authors say “The shifting of splicing position would likely add 8 extra nucleotides from the intron region to the mature mRNA sequence, therefore, could affect the translation process of mRNA into protein”. Does the alteration impact start or stop codons? A schematic view of the predicted effects would be very illustrative.
Besides this major question, I still have the following minor points:
- Page 1, line 1-2. Concerning the change in the title, in my opinion it is not necessary. The original was well
- Page 1 lines 8-9. Instead of “ Until now, MPS I has not been described in detail in Vietnamese” consider the suggestion. Until now, MPS I in Vietnamese has been poorly addressed”.
- Page 2 lines 64,, In the phrase “All of them are Kinh ethenic…..” ethenic is misspelled. And consider this option “All of them are Kinh, the predominant ethnic group in Vietnam, “
- Page 2 lines 72-73. Omit the phase “Hurler syndrome representing the most severe with the earliest onset (1-2 years) and presence of neurocognitive regression, rapid disease progression, hepatosplenomegaly, hernias [9]”. It is too general. Here you must clearly inform whether the 5 patients had the symptoms usually associated with Hurler syndrome
5- Page 4, lines 266-268. Clarify what you meant with the statements “Many of the polymorphisms variants of the sequence have been described in the IDUA gene [23], [7]. The effect of these sequence variants on the functioning of IDUA has not been yet well defined”
Author Response
Dear Sir/Madam,
We are grateful to the reviewer for your valuable suggestions, which have improved the quality of the manuscript. Herein, we addressed them individually:
However, my major concern, which was about c.1190-10C>A, continues. The explanation on the possible deleterious effect of this variation is rather chaotic and unsatisfactory. In page 3 lines 1273-128 the authors write that “…changing from "agccccgcag^ATGAGGAGCA" to "ccccccccag^CCCCGCAGAT", resulting in truncation of the protein” First, it must be more clearly evidenced how that specific splicing disruption will led to a truncated protein. Second, it must be explained how the truncation of the protein arises, given that in lines 264/265 the authors say “The shifting of splicing position would likely add 8 extra nucleotides from the intron region to the mature mRNA sequence, therefore, could affect the translation process of mRNA into protein”. Does the alteration impact start or stop codons? A schematic view of the predicted effects would be very illustrative.
Response: We have already added a schematic of the predicted effects and explained how the variant leads to a truncated protein. The c.1190-10C>A variant may cause activation of an alternative cryptic splice site would likely add 8 extra nucleotides from the intron region to the mature mRNA sequence, resulting in a frameshift which gains a stop codon after 45 amino acids downstream of the mutation site, leads to truncated 212 amino acids
1. Page 1, line 1-2. Concerning the change in the title, in my opinion it is not necessary. The original was well
Response: According to your suggestion, we have recovered the title.
2. Page 1 lines 8-9. Instead of “ Until now, MPS I has not been described in detail in Vietnamese” consider the suggestion. Until now, MPS I in Vietnamese has been poorly addressed”.
Response: According to your suggestion, we have changed “Until now, MPS I has not been described in detail in Vietnamese” to “Until now, MPS I in Vietnamese has been poorly addressed”
3. Page 2 lines 64, In the phrase “All of them are Kinh ethenic…..” ethenic is misspelled. And consider this option “All of them are Kinh, the predominant ethnic group in Vietnam"
Response: According to your suggestion, we have changed “All of them are Kinh ethenic” to “All of them are Kinh, the predominant ethnic group in Vietnam”
4. Page 2 lines 72-73. Omit the phase “Hurler syndrome representing the most severe with the earliest onset (1-2 years) and presence of neurocognitive regression, rapid disease progression, hepatosplenomegaly, hernias [9]”. It is too general. Here you must clearly inform whether the 5 patients had the symptoms usually associated with Hurler syndrome
Response: According to your suggestion: we have removed this phase. We have confirmed the 5 patients had the symptoms of Huler syndrome in the results: “A total of 5 patients were diagnosed with Hurler syndrome”
5. 5- Page 4, lines 266-268. Clarify what you meant with the statements “Many of the polymorphisms variants of the sequence have been described in the IDUA gene [23], [7]. The effect of these sequence variants on the functioning of IDUA has not been yet well defined”
Response: We have already rewritten the statements more clearly.
We would like to send you the newly revised manuscript in the attached file.
Kind Regards,
Bich Ngoc
